# Exploring the Frozen Armory: Antiphage Defense Systems in Cold-Adapted Bacteria with a Focus on CRISPR-Cas Systems

**DOI:** 10.3390/microorganisms12051028

**Published:** 2024-05-20

**Authors:** Greta Daae Sandsdalen, Animesh Kumar, Erik Hjerde

**Affiliations:** Department of Chemistry, UiT the Arctic University of Norway, 9019 Tromsø, Norway; greta.sandsdalen@uit.no (G.D.S.); animesh.kumar@uit.no (A.K.)

**Keywords:** antiphage defense systems, antiviral defense systems, cold-adapted bacteria, CRISPR-Cas, dXTPases, psychrophiles, psychrotolerants

## Abstract

Our understanding of the antiphage defense system arsenal in bacteria is rapidly expanding, but little is known about its occurrence in cold-adapted bacteria. In this study, we aim to shed light on the prevalence and distribution of antiphage defense systems in cold-adapted bacteria, with a focus on CRISPR-Cas systems. Using bioinformatics tools, Prokaryotic Antiviral Defense LOCator (PADLOC) and CRISPRCasTyper, we mapped the presence and diversity of antiphage defense systems in 938 available genomes of cold-adapted bacteria from diverse habitats. We confirmed that CRISPR-Cas systems are less frequent in cold-adapted bacteria, compared to mesophilic and thermophilic species. In contrast, several antiphage defense systems, such as dXTPases and DRTs, appear to be more frequently compared to temperate bacteria. Additionally, our study provides Cas endonuclease candidates with a potential for further development into cold-active CRISPR-Cas genome editing tools. These candidates could have broad applications in research on cold-adapted organisms. Our study provides a first-time map of antiphage defense systems in cold-adapted bacteria and a detailed overview of CRISPR-Cas diversity.

## 1. Introduction

Cold environments cover large proportions of Earth’s area, from polar regions to deep-sea trenches. They harbor unique microbial communities that have adapted to survive and thrive despite challenges such as extreme temperatures, low nutrient availability, and high salinity [1]. Organisms that inhabit cold environments are commonly classified into two overlapping groups: psychrophiles and psychrotolerants (or psychrotrophs). Psychrophiles have an optimal growth temperature of around 15 °C and maximum growth temperature of 20 °C, while psychrotolerants grow optimally around 20 °C and have a maximum growth temperature of 30 °C [2,3]. Psychrophiles predominate in marine ecosystems, whereas bacteria isolated from cold terrestrial environments are most often found to be psychrotolerant [4]. Here, we employ the term ‘Cold-adapted bacteria’ referring to both psychrophilic and psychrotolerant bacteria.

Cold-adapted bacteria have evolved unique mechanisms to overcome challenges such as reduced enzyme activity, protein cold denaturation, decreased membrane fluidity, and intracellular ice formation [5]. Bacterial communities in polar and marine ecosystems are reported to have high levels of viral infection, where the phage threat is especially dominant and responsible for killing approximately 20% of the marine microbial biomass daily [6,7,8].

Bacteria encode multiple lines of defense against phages. These include defense systems with well-studied mechanisms, such as abortive infection (Abi) systems, restriction-modification (RM) systems, and CRISPR-Cas systems, as well as many recently discovered systems with lesser known or unknown modes of action [9,10,11]. Generally, bacteria are known to resist phage infections by blocking phage adsorption and injection, targeting the phage nucleic acids via degradation or synthesis inhibition, or by death by suicide upon phage infection [11,12,13,14]. The latter mode of protection, termed abortive infection (Abi) induces cell death in infected cells, preventing the spread of the virus to other cells in the population. The Abi system is abundant across bacterial genomes and includes diverse mechanisms of action [13].

RM systems are also a widely distributed bacterial defense mechanism, consisting of two key components: restriction enzymes and modification enzymes. This dual system provides a two-pronged defense by recognizing and targeting foreign DNA while safeguarding the host genome [15]. The CRISPR-Cas system also recognizes and targets foreign nucleic acids but is distinguished by its ability to readily acquire new specificities. CRISPR-Cas confers defense against invading genetic elements by integrating short fragments of foreign DNA into the CRISPR array, termed spacers. This integration enables subsequent identification and degradation of complementary nucleic acids (protospacers) [16]. Notably, the system displays intriguing variations in distribution among taxa and environments [17,18,19,20,21]. Although bacterial antiphage defense systems have distinct features and mechanisms, they coexist and complement each other in providing an efficient defense against foreign genetic material in a synergic manner [22,23].

The CRISPR-Cas system has gained a lot of attention in the last decade due to its conversion into a groundbreaking tool for genetic engineering and genome editing, owing to its programmable RNA-guided endonuclease activity [24,25]. Classifications of CRISPR-Cas systems are based on effector complexes, yielding two primary classes and six distinct types: Class 1 (including types I, III, and IV) and Class 2 (including types II, V, and VI). Furthermore, these types can be divided into at least 34 subtypes [19,20,26,27,28]. A fundamental divergence between Class 1 and Class 2 lies in their utilization of multi-Cas effector complexes and single effector nucleases, respectively, which renders Class 2 systems particularly suitable for genome editing and genome engineering applications [28]. For more detailed information on mechanisms and applications for the various CRISPR systems in genome editing and engineering, see reviews by Nishiga and coworkers [29] and Hillary and Ceasar [30]. In addition to CRISPR-Cas, other antiphage defense systems have also been converted into genome editing tools, such as prokaryotic Argonautes (pAgos) and bacterial retrons [31,32].

Antiphage defense systems, including CRISPR-Cas, are known to be widespread in bacteria and the arsenal of unique systems is rapidly expanding [33,34,35]. However, little is known about their occurrence and distribution in cold-adapted bacteria. Previous research has shown that temperature range is a strong predictor of CRISPR-Cas incidence, with increasing abundance observed with rising temperatures [36,37]. Yet, these studies have been limited by a small sample size of cold-adapted bacteria and analyzes of CRISPRs and Cas clusters individually. The prevalence and distribution of other antiphage defense systems in cold-adapted bacteria remain unexplored.

In this study, we aimed to map the prevalence and distribution of antiphage defense systems in cold-adapted bacteria from various cold environments. We constructed a dataset of high quality, assembled bacterial genomes collected from different habitats, including cold marine waters, sea ice, glaciers, and permafrost. Bioinformatic tools were applied to predict antiphage defense systems in the genomes of these microorganisms. Our study provides an overview of the mechanisms that cold-adapted bacteria have evolved to defend themselves against viral infections in cold environments. Additionally, we provide a detailed map of the diversity and distribution of CRISPR-Cas systems in cold-adapted bacteria, which has not been previously established.

## 2. Materials and Methods

A dataset of cold-adapted bacterial genome sequences was generated based on the five databases MarRef v.1.7, MarDB v.1.6, Ocean Microbiomics Database (OMD) v.1.1, BacDive (https://bacdive.dsmz.de/ (accessed on 23 January 2023)), and TEMPURA (http://togodb.org/db/tempura (accessed on 4 April 2023)). These databases, combined, hold a comprehensive collection of bacterial genomes from diverse cold environments.

Filtering from MarDB, MarRef, and OMD databases was primarily focused on isolation location. To exclude mesophilic and thermophilic species, the entries were filtered based on their geographical isolation data. Here, we selected genomes from bacteria only isolated above 60° N and below 50° S, where ocean surface temperatures are <10 °C [38]. All genomes from MarRef and MarDB bacteria classified as mesophilic, thermotolerant, and thermophilic were excluded, as well as those bacteria isolated from homoiotherms and hydrothermal vents. Genomes from BacDive and TEMPURA were filtered based on experimental temperature data. Genomic data from BacDive were collected 23 January 2023 through their advanced search functions using the following filters: Temperature range: psychrophilic; Test result (temperature): positive; Genome seq. accession number: (contains) GCA; and Sample type/isolated/NOT contains: marine. The results were further filtered manually. The criteria for exclusion were optimum growth temperature > 25 °C, conflicting experimental data for growth temperature, isolated from hot environments/locations, and missing sequencing data. A total of 61.48% of hits were removed during manual filtration. Genomic data from TEMPURA were collected on 4 April 2023. Inclusion criteria: optimum growth temperature ≤ 20 °C; maximum growth temperature < 30 °C; and registered accession number.

The genome sequence metadata were evaluated using standard quality control measures developed by the Genomic Standards Consortium (GSC) [39] prior to download and analysis to ensure data quality and consistency.

All single amplified genomes (SAGs) were excluded from the dataset, due to contamination challenges associated with whole genome amplification techniques [40]. All remaining genomes, including the metagenome-assembled genomes (MAGs) from MarDB and OMD databases, were quality assessed using CheckM (v1.2.2) [41]. Low-quality sequences with low completeness (<90%) or high contamination (>5%) were excluded from the dataset. In addition, accession ID duplicates across the five databases were identified and removed. All genomes in the dataset were taxonomically reclassified using the GTDB-Tk (v2.3.0) workflow against the Genome Taxonomy Database (GTDB) r.214. Genomes classified as archaea by GTDB were also excluded (N = 11).

The genomes of interest from MarRef (N = 86), MarDB (N = 225), OMD (N = 299), BacDive (N = 305), and TEMPURA (N = 25) were downloaded from the respective databases using their provided application programming interfaces (APIs) or download portals. A total of 938 genomes were downloaded from the five databases.

All genomes in the dataset were annotated for CRISPR-Cas systems (gene clusters and arrays) using CRISPRCasTyper (cctyper v1.8.0) [42] at the default parameter [subtype probability above 0.754]. Only the CRISPRs part of an intact CRISPR-Cas loci were included. Other prokaryotic antiphage defense systems were predicted using the Prokaryotic Antiphage Defense LOCator: PADLOC (v1.1.4) [33] using parameter a E-value < 0.01 and coverage > 0.8, where it assigned a unique system number referring to each system identified in a genome.

For downstream analysis, the presence of each defense system was counted based on the system number in each genome. All PADLOC-predicted defense system subtypes were collapsed to type level and their prevalence and phylogenetic distribution were analyzed in R v3.6.2. Phylogenetic trees were plotted using the phyloseq v1.44.0 and ggtree v3.8.2 R packages [43,44]. Principal component analysis (PCA) was performed to examine possible correlations between the low prevalence of CRISPR-Cas systems against the high prevalence of other systems, such as dXTPases. PCA analysis was performed at the taxonomic rank “family” using the factoMineR v2.8 [45] and factoextra v1.0.7 [46] packages in R.

## 3. Results

### 3.1. Dataset of 938 Cold-Adapted Bacterial Genomes

To map antiphage defense systems in cold-adapted bacteria, we generated a dataset of assembled genomes of bacteria that were considered cold-adapted either based on isolation location (MarRef, MarDB, and OMD databases) or experimental temperature growth data (BacDive and TEMPURA databases). The final dataset includes 938 bacterial genomes collected from diverse habitats, including marine waters, sea ice, glaciers, and permafrost. The table of genomes in the dataset is given in Appendix A. In the downloaded dataset, we observed a predominance of the *Pseudomonadota* (or *Proteobacteria*), *Bacteriodota* (or *Bacteroidetes* or FCB group bacteria), and *Actinomycetota* (or *Actinobacteria*) phyla, with fewer representatives from the *Bacillota* (or *Firmicutes*), *Planctomycetota* (or *Planctobacteria*), and *Verrucomicrobiota* phyla. The taxonomic distribution of our dataset is shown in Appendix A.

### 3.2. Abundance and Distribution of Antiphage Defense Systems in Cold-Adapted Bacteria

The bioinformatics tools Prokaryotic Antiviral Defense LOCator (PADLOC) and CRISPRCasTyper were applied to the dataset of cold-adapted bacterial genomes to predict antiphage defense systems. PADLOC analysis predicted the presence of antiphage defense systems in 799 genomes, identifying 48 unique systems and a total of 145 system subtypes—see Appendix A. Additionally, CRISPRCasTyper, which utilizes the machine learning approach, identified the presence of CRISPRs and Cas operons (20 subtypes) in 166 genomes—see Appendix A. Among the antiphage systems illustrated in Figure 1, RM systems were found to be the most dominant (~78% genomes contained these systems) in cold-adapted bacteria. Surprisingly, dXTPases (dCTP deaminases and dGTPases combined) had the second highest prevalence, found to be present in 44% of the genomes, closely followed by the Abi systems (42%). Subsequently, Defense-associated reverse transcriptase (DRT) was found in ~25% of the genomes in our dataset and CBASS, Gabija, and CRISPR-Cas were found to be equally dominant (~17%) in our dataset.

A more detailed overview of prevalence linked to phylogenetic distribution of the identified antiphage systems is presented in Figure 2. This analysis shows that some antiphage systems are frequently found across the phylogenetic groups, such as RM and Abi. On the other hand, dXTPases, CRISPR-Cas, and the anti-plasmid system Wadjet show an unequal frequency across phylogenetic groups. Prevalence linked to phylogenetic distribution at the family level is presented in Appendix A.

### 3.3. CRISPR-Cas Systems in Cold-Adapted Bacteria

The CRISPR-Cas systems were found to be present in 17.7% of the genomes in the dataset (Figure 3a), and the Class 1 systems were most abundant (Figure 3a,b). Based on type level, Types I, III, and II were most abundant, whereas Types IV, V, VI were almost absent in our dataset (Figure 3c). When analyzing the total number of CRISPR systems predicted in the dataset genomes (Figure 3b), where several genomes had multiple occurrences, we observed that subtype II-C (under Class 2, Type II) was most frequent. Of the Class 1 systems, subtype I-F was most frequently predicted. Interestingly, *Bacteroidota* were harboring more subtype II-C loci, while subtypes I-C, I-E and I-F were frequent in *Pseudomonadota* (Figure 3c).

### 3.4. Candidates for Cold-Active Genome Editing and Genome Engineering

To categorize enzymes with a potential for development into biotechnological tools, we screened for antiphage defense systems harboring effector proteins Cas9, Cas12a, and Cas13 from CRISPR-Cas systems, pAgos, and reverse transcriptases (RTs) from retron systems.

The CRISPRCasTyper analysis predicted fifty-two subtype II-C systems within 48 bacterial genomes, two subtype II-A and one subtype II-B systems (Figure 3b), with corresponding Cas9 effector endonucleases. We also identified two V-A systems (Cas12a effector), two VI-A (Cas13a effector), and six VI-B1 systems (Cas13b effector). Through PADLOC analysis, we identified 51 genes encoding pAgos within our dataset from 47 genomes. The retron-mediated genome editing system (REGES) requires an RT and corresponding non-coding RNA (ncRNA). From PADLOC analysis, we have identified 83 retron RTs from 81 bacterial genomes.

A list of the bacterial genomes encoding these enzymes, as well as their genomic positions, is given in Appendix A).

### 3.5. Patterns in Antiphage Defense System Occurrence

To further investigate the low abundancy of CRISPR-Cas systems in cold-adapted bacteria, correlation analysis was performed to look for patterns in antiphage defense system occurrence. Principal component analysis (PCA) was applied to the combined output data from PADLOC and CRISPRCasTyper (Figure 4). PCA performed for antiphage defense systems, classified at the taxonomic rank family, explained 22.3% of total variation with dimension (dim) 1 and 2. Dim1 was positively associated with all selected systems except dXTPases, while Dim2 was predominantly positively associated except Argonautes, CRISPR-Cas, and Abi.

## 4. Discussion

In this study we aimed to investigate the abundance and diversity of antiphage defense systems in cold-adapted bacteria. In reviewing the current literature on antiphage defense system abundance and distribution, we did not find any study categorizing the result based in bacterial growth temperature. Therefore, we compared our results to studies based on datasets of bacterial genomes in general. Our study shows that cold-adapted bacteria possess a diverse range of antiphage defense systems, with RM, dXTPases, and Abi systems being the most prevalent.

We found dXTPases to be the second most prevalent antiphage defense system in our dataset of cold-adapted bacterial genomes, which is somewhat surprising. In reviewing the literature, Tal and coworkers [47] reported dCTP deaminases in 2.5% and dGTPases in about 6% of the analyzed genomes. A review of antiphage defense systems of bacteria, by Georjon and Bernheim [34], reported the mean number of copies of the dGTPase system encoded in one bacterial genome to be only 0.07. In contrast, we found that 44.2% of the genomes assessed genes encoding dXTPases (including both dCTP deaminases and dGTPases). The dXTPase defense proteins halts phage replication in the bacterial cell by depleting them of specific deoxynucleotides (dCTP or dGTP) in the nucleotide pool, thus starving the phage of essential DNA building blocks. This has shown to be a potent antiphage strategy shared by both prokaryotes and eukaryotes [47]. Furthermore, the taxonomic distribution of dXTPases revealed that dCTP deaminases and dGTPases are predominantly present in the phyla *Pseudomonadota*, with additional occurrences in *Actinomycetota* and *Bacillota*. dGTPases were also found to be occasionally present in *Verrucomicrobiota* [47]. Our findings align with this distribution, and the high occurrence of dXTPases may, in part, be attributed to the high count of *Pseudomonadota* in our dataset. The prevalence of dXTPases in our study highlights the importance of these enzymes in the defense strategy of *Pseudomonadota*.

Defense-associated reverse transcriptase’s (DRTs) were found in 25.6% of the investigated genomes. These systems consist of reverse transcriptase’s (also known as RNA-directed DNA Polymerases) from different unknown groups (Ugs) that act alone, together with small membrane proteins or as two RTs from two different Ugs coupled with a ncRNA [48,49]. DRTs have previously been reported to be of low abundancy in bacterial genomes. Georjon and Bernheim [34] reported the mean number of copies of the DRT system encoded in one bacterial genome to be 0.05 and Tesson and co-workers [50] found DRT systems in 4.6% of assessed bacterial genomes. DRTs have been shown to confer resistance to phages, although through an unknown mechanism.

We confirmed that the presence of CRISPR-Cas systems is less frequent in cold-adapted bacteria, compared to mesophilic and thermophilic bacteria where CRISPR systems are found in 49% and 92% of analyzed genomes, respectively [36]. In our dataset, 17.7% of the genomes contained CRISPR operons, which encompassed gene clusters and CRISPR arrays in close vicinity. Additionally, we identified several orphan Cas gene clusters and orphan CRISPR arrays; however, these were excluded from our reported findings since they do not constitute functional defense systems.

As an adaptive immune system, CRISPR-Cas has obvious theoretical benefits. It effectively targets and degrades repeat invaders with its ‘molecular memory’ (spacers) stored within the CRISPR array. In practice they also have costs that negatively impact the reproduction and survival of their bacterial hosts [21,51]. The CRISPR operon consists of several Cas genes and a CRISPR array of direct repeats and incorporated spacers that bears a metabolic burden in both cellular resources and general bioenergetic demands. CRISPR-Cas systems target not only phages but also a variety of mobile genetic elements (MGEs) including plasmids and integrative conjugative elements. This may affect bacterial adaptation via horizontal gene transfer (HGT) [51]. Cold-adapted bacterial species have been shown to rely on the acquisition of new genetic material through HGT for rapid adaptation to changing environmental conditions [52,53,54,55]. Additionally, a high mutation rate in the target phages, or in the host genome, will lead to a loss of the spacers’ complementary effectiveness [51]. Studies have shown that mutation rates depend on temperature, increasing toward both temperature extremes [56]. With the costs in mind, it would seem beneficial for cold-adapted bacteria to select against CRISPR-Cas systems.

The observed decline in CRISPR-Cas system prevalence with decreasing environmental temperatures has been attributed to the predominance of cellular predation, rather than viral infection, as the primary cause of bacterial mortality in these conditions [37]. Here, we propose an alternate explanation for the low CRISPR-Cas abundance, although we recognize that predator grazing might also influence this trend. Our correlation analysis reveals that dXTPases often occur in the genomes of cold-adapted bacteria where CRISPR-Cas systems are absent, indicating that these are not complimentary defense systems and, in contrast, appear to be mutually exclusive. Hence, it could conceivably be hypothesized that cold-adapted bacteria select against the energy-costly and HGT-hostile CRISPR-Cas systems in favor of the less complex antiphage defense systems, such as dXTPases. Both systems target phage nucleic acids, although through different strategies.

When comparing our findings to those of previous studies on more temperate bacterial species, it must be pointed out that phylogenetic distributions vary amongst psychrophilic, mesophilic, and thermophilic bacterial species. The phylogenetic distribution in our dataset is biased towards the phylum’s *Pseudomonadota* and *Bacteroidota*, contributing 551 genomes of the total 938. This was expected since bacteria from these phylum’s are the most commonly reported microorganisms in deep-sea and polar regions [38]. *Bacillota* and *Deinococcota* are also frequently found in Antarctic and alpine environments. It should also be noted that the findings in this study are somewhat limited by the size of the dataset; therefore, the results might not be representative of all cold-adapted bacteria.

In 2012, CRISPR-Cas was reconstructed into a genome editing tool and, thus, revolutionized the field of biotechnology [24]. The CRISPR technology gained popularity for its simplicity, affordability, and specificity. The CRISPR tools, at present, are mostly from mesophilic species, making genome editing in cold-living organisms, such as poikilotherms, challenging. Cold-active Cas enzymes could solve these issues, providing an optimal temperature activity aligned with the growth temperature of the targeted cells or organism. The heat-labile nature of cold-active enzymes could also provide an advantage in tightly controlling genome editing with temperature. However, it is not directly implied that genes from cold-adapted bacteria encode cold-active proteins [57]. To determine the optimal enzymatic temperature the proteins must be recombinantly expressed and tested. Here, we have identified fifty-five Cas9 genes from fifty-one bacterial genomes and two Cas12a genes with potential as cold-active genome editing tools. Additionally, we identified two Cas13a and six Cas13b genes that have the potential to be cold-active RNA editors and/or RNA screening tools.

Further interesting findings are the prokaryotic argonautes (pAgos) and retrons from cold-adapted bacteria, which also have potential as cold-active genome editing tools. Like Cas endonucleases, pAgos are easily programmed to cleave DNA and/or RNA targets. Most pAgos have short, single-stranded DNA guides, which are more stable than RNA guides, and they also present other advantages over Cas such as no sequence restrictions and compact size [32]. To date, only one psychrotolerant pAgo (MbpAgo) has been experimentally validated [32,58]. Here, we have identified 51 candidates from both psychrophilic and psychrotolerant bacterial hosts. In contrast to CRISPR-Cas and pAgos, bacterial retrons confer defense against phages via abortive infection with the defensive unit composed of ncRNA, RT, and an effector protein [59]. Recently, retron ncRNA and RT have been utilized in the versatile retron-mediated genome editing system (REGES), providing promising results for efficient genome editing in prokaryotes [31]. Here, we have identified 83 retron RTs with potentially cold-active properties.

## 5. Conclusions

Our study provides a comprehensive overview of the prevalence and distribution of antiphage defense systems in cold-adapted bacteria from various cold environments. RM and Abi systems are widely distributed among cold-adapted bacteria, as for bacteria in general. We have also confirmed that CRISPR-Cas systems are less prevalent in bacterial species located in cold environments. Interestingly, we found high prevalence of DRT and dXTPases systems, suggesting that cold-adapted bacteria select against CRISPR-Cas systems in favor of other antiphage defense systems. The findings from this study also offer potential applications in biotechnology by identifying candidates of Cas endonucleases, pAgo endonucleases, and retron RTs, which can be further characterized and developed as cold-active genome editing and/or engineering tools.

## Figures and Tables

**Figure 1 microorganisms-12-01028-f001:**
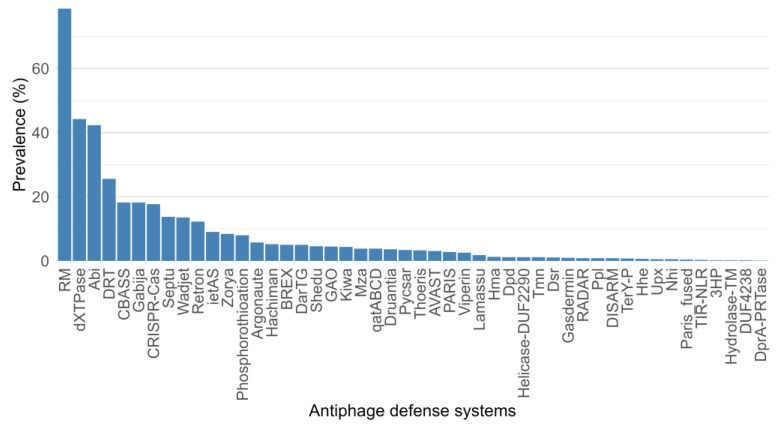
Prevalence, in percentage, of genomes encoding the 48 predicted antiphage defense systems.

**Figure 2 microorganisms-12-01028-f002:**
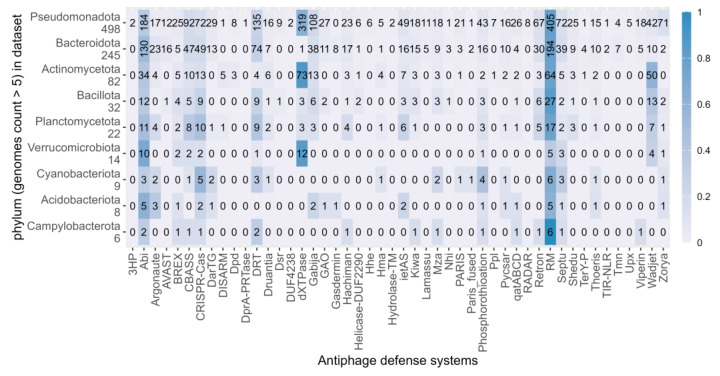
Distribution of antiphage defense systems across the prokaryotic phylum. Only phylogenetic groups with more than 5 genomes are represented. The number of genomes in the dataset is represented under each phylum’s name. The heatmap illustrates the prevalence of fractional occurrence of each antiphage defense system within each phylogenetic group (per row), with a color legend on the right. The absolute number of genomes encoding a particular system is specified in each cell.

**Figure 3 microorganisms-12-01028-f003:**
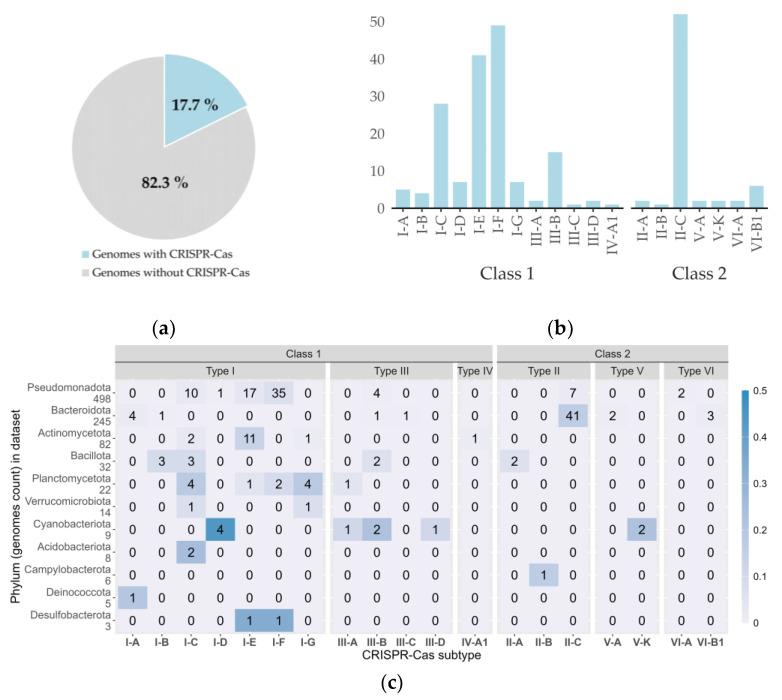
(**a**) Percentage of cold-adapted bacterial genomes harboring CRISPR-Cas systems. (**b**) Distribution of CRISPR-Cas subtypes identified in the dataset. *Y*-axis gives the total count of predicted systems. (**c**) Distribution of CRISPR-Cas systems per prokaryotic phylum. Each row represents a phylum, and the number of genomes in the dataset is shown next to each phylum’s name. The heatmap illustrates the CRISPR subtype frequency within each phylum, with a color legend on the right. The absolute number of genomes encoding a particular CRISPR subtype is specified in each cell.

**Figure 4 microorganisms-12-01028-f004:**
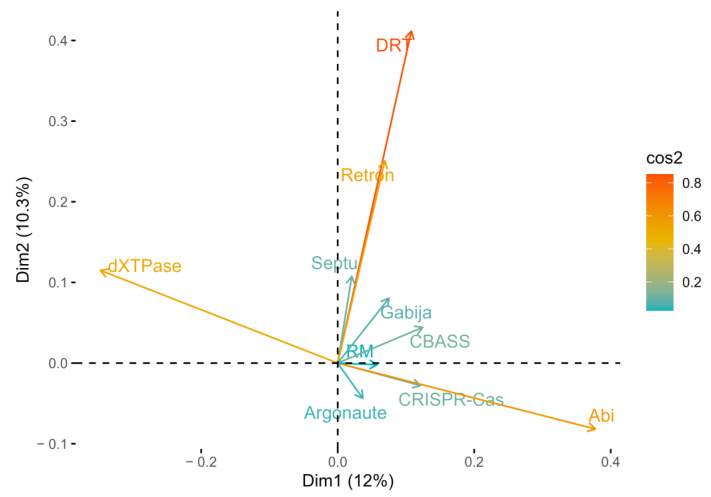
Principal components analysis (PCA) of antiphase system variables for bacterial family. Arrows corresponding to 10 antiphage system variable contributions (cos2 > 0.02) are shown.

## Data Availability

The original contributions presented in the study are within the article/Appendix A and on a GitHub repository at https://github.com/Animesh911/Antiphage-defense-system (accessed on 14 May 2024). Further inquiries can be directed to the corresponding author/s.

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
