# Peer review of "Exploring the Frozen Armory: Antiphage Defense Systems in Cold-Adapted Bacteria with a Focus on CRISPR-Cas Systems"

_microorganisms, 2024, doi:10.3390/microorganisms12051028_

Round 1
Reviewer 1 Report
Comments and Suggestions for Authors
The manuscript "Exploring the frozen armory: Antiphage defense systems in cold-adapted bacteria with focus on CRISPR-Cas systems" by G.D. Sandsdalen, A. Kumar, E. Hjerde is devoted to the evaluation of antiphage defence system in psychrophilic and psychrotolerant bacteria according the public genomic data. The authors constructed the dataset consisting 938 relevant genomes. All genomes were validated and analyzed to identify known antifage defence systems.
I guess the work is interesting assessment with important environmental ecological implications. Despite the "pure bioinformatic" character I recommend to accept the work after minor revision. The corrections needed are listed below:
line 5: Affiliation 1; - the example text should be deleted;
line 263: Actinomycedota - Actinomycetota;
line 357: The description of supporting info should be more informative. Please, add more detailed description and/or reorganize the supporting files.
For example:
Supplementary Figure S1: Phylogenetic distribution of dataset.
Supplementary Figure S2: Distribution of antiphage defense systems across prokaryotic family.
It's much more informative and easy for navigation.
line 380: Some references include repeated text "doi:". Please, check refs. 2, 15, 16, 24
Comments on the Quality of English LanguageSee the section above
Author Response
Dear Reviewer 1,
Thank you for taking the time to assess our manuscript.
R1.1) line 5: Affiliation 1; - the example text should be deleted
R R1.1) Thank you for pointing this out. The example text on line 5 has been removed.
R1.2) line 263: Actinomycedota – Actinomycetota
R R1.2) Thank you for pointing out this error. In line 262 “Actinomycedota” has been replaced with “Actinomycetota”.
R1.3) line 357: The description of supporting info should be more informative. Please, add more detailed description and/or reorganize the supporting files.
R R1.3) We agree with this. We have, accordingly, changed the description of the supplementary materials to make it more informative, lines 356-367. A minor reorganization of the supplementary files was also done. Table S1 in Supplementary File 2 was removed, as the table contents are repeated in Supplementary File 3. We also discovered errors regarding supplementary table names. In line 157 “Table S2” has been replaced with “Table S1”. In line 226 “Tables S1, S2 and S3” has been replaced with “Tables S2, S3 and S4”.
R1.4) line 380: Some references include repeated text "doi:". Please, check refs. 2, 15, 16, 24
R R1.4) Thank you for pointing out this error. We have now removed the repeated “doi:” in references 2, 15, 16 and 24.
Best Regards,
Ms. Greta Daae Sandsdalen,
Mr. Animesh Kumar,
Dr. Erik Hjerde
Reviewer 2 Report
Comments and Suggestions for Authors
1- Aim, novelty and significance
· This study aimed to shed light on the prevalence and distribution of the anti-phage defense systems in cold-adapted bacteria, with a focus on CRISPR-Cas systems. Hence the study provided novel and useful information in that respect including a detailed overview of CRISPR-Cas diversity.
· Accordingly, the study confirmed that CRISPR-Cas systems are less frequent in cold-adapted bacteria compared to mesophilic and thermophilic species. ِ
· Additionally, the study pinpointed Cas endonuclease candidates with a potential broad applications in research on cold-adapted organisms.
· This work is novel and significant and can inspire for other useful discoveries.
2-The title
The title should be better directly expressing the content and objective of the article. It may be slightly modified into: [Exploring and mapping antiphage defense systems in cold-adapted bacteria with a focus on CRISPR-Cas systems].
3- Illustrations
Figure 1:
-Expand the length of x-axis by about 10% to be easily legible.
-There is no need to explain the title of x-axis in the figure title.
-Just refer to the definition of the prevalence under the figure.
Figures 2-4 should be enlarged even through using landscape desing to be easily legible and communicating.
4- Minor corrections
L8, change (but still, little is known ) to (but little is known )
L8, change (about their occurrence) to (about its occurrence)
L15-16, change the sentence into (In contrast, several anti-phage defense systems, such as dXTPases and DRTs, appear more frequently in temperate bacteria.)
L26-27, change into (Cold environments cover much of the earth's area, from polar regions to deep-sea trenches.)
L31, change (temperature around 15°C) to (temperature of around 15°C)
L33, change (have maximum growth) to (have a maximum growth)
L107, change (bacteria classified as mesophilic) to (bacteria, classified as mesophilic)
L111, change (with following) to (with the following)
L119-121, change into (The genome sequence metadata were evaluated using standard quality control measures developed by the Genomic Standards Consortium (GSC) [39] prior to download and analysis to ensure data quality and consistency.)
L142, change (For downstream analysis the) to (For downstream analysis, the)
L143, change (on system number) to (on the system number)
L152, change (bacteria we ) to (bacteria, we )
L157, change (Table of) to (The table of)
L158, change (dataset we) to (dataset, we)
L164, change (The bioinformatic tools) to (The bioinformatics tools ).
L165, change (CRISPRCasTyper were applied) to (CRISPRCasTyper was applied)
L168-169, change (utilizes machine) to (utilizes the machine)
L187, change (at family level) to (at the family level)
L215, change (with potential ) to (with a potential )
L215, change (tools we ) to (tools, we )
L222 and L225, change (analysis we) to (analysis, we)
L302, change (CRISPR-Cas abundancy) to (CRISPR-Cas abundance)
L306, change (in contrast appears ) to (in contrast, appear )
L307, change (energy costly ) to (energy-costly )
Comments on the Quality of English Language
Only minor editing is required
Author Response
Dear Reviewer 2,
Thank you for taking the time to assess our manuscript.
R2.1) This work is novel and significant and can inspire for other useful discoveries.
R R2.1) Thank you. We hope this work can inspire useful discoveries.
R2.2) The title should be better directly expressing the content and objective of the article. It may be slightly modified into: [Exploring and mapping antiphage defense systems in cold-adapted bacteria with a focus on CRISPR-Cas systems].
R R2.2) Thank you for this suggestion. We have discussed the title change but decided to keep the original title. Searching for “exploring and mapping” on Google Scholar gave over 3700 hits, while “exploring the frozen” gave 71 hits. We believe the current title may increase the visibility of the research article.
R2.3) Figure 1: -Expand the length of x-axis by about 10% to be easily legible. -There is no need to explain the title of x-axis in the figure title. -Just refer to the definition of the prevalence under the figure.
R R2.3) Thank you for the suggestions. The length of the x-axis in Figure 1 has been extended by 10%. Figure 1 figure text has been changed to “Prevalence, in percentage, of genomes encoding the 48 predicted antiphage defense systems.” (line 179-180).
R2.4) Figures 2-4 should be enlarged even through using landscape desing to be easily legible and communicating.
R R2.4) To improve legibility, the font size in the figures has been increased where possible. In Figure 2, numbers within the heatmap are increased. x/y axis title was not increased due to space restrictions within the figure. The legend scale has been improved by including top value. In Figure 3a, the text font size has been increased. In Figure 3b, x/y axis and the underlying text font size have been increased. In Figure 3c, numbers inside heatmap, legend values, and axis title size are now increased. X-axis text type has been changed to bold. We explored rotating the axis of Figures 2 and 3c. In our opinion, this did not improve the legibility.
In addition to the above comments, all minor corrections regarding spelling and grammar have been corrected. We thank the reviewer for pointing out grammatical errors and appreciate the suggestions to improve the language.
Best Regards,
Ms. Greta Daae Sandsdalen,
Mr. Animesh Kumar,
Dr. Erik Hjerde
Reviewer 3 Report
Comments and Suggestions for Authors
In this manuscript, the authors present an overview of the occurrence of phage resistance in cold-adapted bacteria using a large database of bacteria from all parts of the world.
I strongly suggest that the authors improve the figures. Figures 1-2-3 are difficult to read, and the text added next to each figure is almost completely illegible.
The text alternates between 'cold-adapted bacteria' and 'psychrophiles' and 'psychrotolerants' without a clear distinction or definition, which may confuse the reader.
Author Response
Dear Reviewer 3,
Thank you for taking the time to assess our manuscript.
R3.1) I strongly suggest that the authors improve the figures. Figures 1-2-3 are difficult to read, and the text added next to each figure is almost completely illegible.
R R3.1) Thank you for this concrete feedback. To improve the legibility of figures 1-2-3, we have increased the text font size where possible and the x-axis of Figure 1 has been extended by 10%. To improve legibility, the font size in the figures has been increased where possible. In Figure 2, numbers within the heatmap are increased. x/y axis title was not increased due to space restrictions within the figure. The legend scale has been improved by including top value. In Figure 3a, the text font size has been increased. In Figure 3b, x/y axis and the underlying text font size have been increased. In Figure 3c, numbers inside the heatmap, legend values, and axis title size are now increased. X-axis text type has been changed to bold. We explored rotating the axis of Figures 2 and 3c. In our opinion, this did not improve the legibility.
R3.2) The text alternates between 'cold-adapted bacteria' and 'psychrophiles' and 'psychrotolerants' without a clear distinction or definition, which may confuse the reader.
R R3.2) We consider the distinction and definition made in lines 29-36 as sufficient:
“Organisms that inhabit cold environments are commonly classified into two overlapping groups: psychrophiles and psychrotolerants (or psychrotrophs). Psychrophiles have an optimal growth temperature of around 15°C and maximum growth temperature of 20°C, while psychrotolerants grow optimally around 20°C and have a maximum growth temperature of 30°C [2,3]. Psychrophiles predominate in marine ecosystems, whereas bacteria isolated from cold terrestrial environments are most often found to be psychrotolerant [4]. Here, we employ the term ‘Cold-adapted bacteria’ referring to both psychrophilic and psychrotolerant bacteria.”
We agree with the reviewer that alternating between the above terms may confuse. Accordingly, we have changed “potentially psychrophilic properties” to “potentially cold-active properties” in line 344. Here, we have chosen “cold-active” as opposed to “cold-adapted”, as it is more frequently used to describe an enzymatic property.
Best Regards,
Ms. Greta Daae Sandsdalen,
Mr. Animesh Kumar,
Dr. Erik Hjerde